# Associations between damage location and five main body region injuries of MAIS 3–6 injured occupants

Youming Tang,[1,2,3] Libo Cao,[2] Steven Kan[3]

For numbered affiliations see end of article.

**Correspondence to**
Dr Youming Tang;
tangyouming@xmut.edu.cn

## ABSTRACT

**Objectives:** To examine the damage location distribution of five main body region injuries of maximum abbreviated injury score (MAIS) 3–6 injured occupants for nearside struck vehicle in front-to-side impact crashes.

**Design and setting:** MAIS 3–6 injured occupants information was extracted from the US-National Automotive Sampling System/Crashworthiness Data System in the year 2007; it included the head/face/neck, chest, pelvis, upper extremity and lower extremity. Struck vehicle collision damage was classified in a three-dimensional system according to the J224 Collision Deformation Classification of SAE Surface Vehicle Standard.

**Participants:** Nearside occupants seated directly adjacent to the struck side of the vehicle with MAIS 3–6 injured, in light truck vehicles–passenger cars (LTV–PC) side impact crashes.

**Outcome measures:** Distribution of MAIS 3–6 injured occupants by body regions and specific location of damage (lateral direction, horizontal direction and vertical direction) were examined. Injury risk ratio was also assessed.

**Results:** The lateral crush zone contributed to MAIS 3–6 injured occupants (n=705) and 50th centile injury risks when extended into zone 3. When the crush extended to zone 4, the injury risk ratio of MAIS 3–6 injured occupants approached 81%. The horizontal crush zones contributing to the highest injury risk ratio of MAIS 3–6 occupants were zones 'D' and 'Y', and the injury risk ratios were 25.4% and 36.9%, respectively. In contrast, the lowest injury risk ratio was 5.67% caused by zone 'B'. The vertical crush zone which contributed to the highest injury risk ratio of MAIS 3–6 occupants was zone 'E', whose injury risk ratio was 58%. In contrast, the lowest injury risk ratio was 0.14% caused by zone 'G+M'.

**Conclusions:** The highest injury risk ratio of MAIS 3–6 injured occupants caused by crush intrusion between 40 and 60 cm in LTV–PC nearside impact collisions and the damage region of the struck vehicle was in the zones 'E' and 'Y'.

## INTRODUCTION

Since 1980, there has been an increase in fatalities resulting from the collision of passenger cars (PC) and light truck vehicles (LTV), because LTV have became the vehicle of choice on the road, resulting in a great deal of

## Strengths and limitations of this study

- This study provides better understanding of struck vehicle characteristics and crash conditions in the current and future side crash environment.
- This research shows that the 50th centile risk of MAIS 3–6 injuries to front-to-side occupants between LTV and PC corresponds to the extent of damage in zone 3, specific horizontal location of damage in zone 'Y' and specific vertical location of damage in zone 'E'.
- This study suggests considering head, chest and pelvis injury prevention a priority to reduce the risk of deaths and fatal injuries in MAIS 3–6 injured occupants.
- The injury risk ratios of five main body regions of MAIS 0–2 injured occupants were excluded.

vehicle incompatibility between the body structures of LTV and PC.[1–4] LTV include sport utility vehicles (SUV), pickup trucks and vans; all have the same truck frames and are weighted no more than 2014 kg.[5]

The reinforcement of PC is most commonly a single beam at the mid-door or lower door cage, leaving the upper portion of the door unprotected. When LTV collide with PC, the higher bumper frame contacts the door above the reinforcement, creating upper thoracic and head injuries.[6 7] Hence, vehicle mismatch between PC and LTV is associated with deaths and serious injuries of occupants in automotive crashes.[8] In the present study, the US national database was used to assess the nearside impact injury risks of five main body regions of maximum Abbreviated Injury Score (MAIS) 3–6 injured occupants between PC and LTV.

## MATERIALS AND METHODS
### Database of the National Automotive Sampling System/Crashworthiness Data System

The objective was to estimate the injury risks of MAIS 3–6 injured occupants by extent of

damage to the impacted vehicle in front-to-side impact crashes using real-world crash cases extracted from the database of the National Automotive Sampling System/Crashworthiness Data System (NASS/CDS). NASS/CDS is a detailed representative sample of motor vehicle crashes that occur within the USA. Each year roughly 5000 detailed crash investigations are conducted by NASS investigators. Each case in the sample is assigned a weighting factor, which is equivalent to the inverse probability of such a vehicle incident occurring in a given year. Crashes that occurred in the year 2007 were selected; the number of crash cases was 4963 and after weighted was 2 454 014 as collected by the National Highway Traffic Safety Administration.

## Selection criteria

In order to extract relevant data from the NASS/CDS datasets, the following selection criteria were established:

▶ Cases in which only two vehicles were involved in the collision.
▶ Cases that resulted in a vehicle rollover were excluded.
▶ Crash configuration involved a front-to-side impact.
▶ Both of the vehicles involved could be classified as PC and LTV including a mini-van, SUV, van and light pickup truck.
▶ Striking vehicle was defined as an impact which resulted in the primary general area of damage being located at the front of the vehicle; it was also defined as having a direction of force between 330° and 30° (11'o clock to 1'o clock).
▶ Struck vehicle was defined as an impact which resulted in the primary general area of damage being located on either the right or the left side of the vehicle; it was also defined as having either a direction of force between 30° and 150° (1'o clock to 5'o clock right side impact) or between 210° and 330° (11'o clock to 7'o clock left side impact).
▶ Injury analysis was conducted based on those injuries sustained by the nearside occupants seated directly adjacent to the struck side of the vehicle, in an LTV–PC side impact crash.

## Collision deformation classification

According to the 'J224 Collision Deformation Classification' of SAE Surface Vehicle Standard, the extent of residual deformation induced by a vehicle was represented from the horizontal, lateral and vertical directions.[9] For passenger cars, the extent of residual deformation in lateral direction is classified using a nine-zone extent system. The extent of damage recorded is dependent on the maximum cross-section which the deformation extends to within a vehicle as a result of an impact. This measure can also be used to indicate the severity of impact of a vehicle.

A vehicle's horizontal areas to be used in locating the deformation for three independent extents and three composite extents are 'P', 'F', 'B' and 'Y', 'Z', 'D',

**Table 1** Investigation of road vehicle crash types in the USA in 2007

| Crash type | Weighted cases | Weighted occupants (not including pedestrians) |
|---|---|---|
| Single vehicle | 832 385 (33.9%) | 1 198 553 (19.9%) |
| Vehicle to vehicle | 1 361 888 (55.5%) | 3 753 285 (62.2%) |
| Multiple vehicle | 259 741 (10.6%) | 1 082 825 (17.9%) |
| Total | 2 454 014 (100%) | 6 034 663 (100%) |

respectively. 'F' and 'B' are side deformation areas forward and rearward of 'P', respectively.

The vertical location of deformations associated with impacts are classified as 'A', 'E', 'G', 'M' or 'L'. The extent zones 'A' and 'E' are used for a vehicle with both side deformations located at the rear end, while 'G', 'M' and 'L' are located at the front end. Zone 'A' is used to classify impacts where the vehicle deformation resulted from an overhanging structure shaped like an inverted step in which the vertical surfaces are at least 760 mm apart.[9]

## RESULTS
### PC–LTV crashes

The information presented in table 1 shows that about 56% of road vehicle crashes resulted from a vehicle-to-vehicle impact. The vehicle-to-vehicle impact crashes caused about 62% injuries; this figure relates to occupants of the vehicles and excludes the pedestrians. As can be seen, vehicle-to-vehicle crashes result in more serious injuries probability per collision than single-vehicle crashes and multiple-vehicle crashes.

Table 2 shows the injury risks ratio of MAIS 3–6 injured occupants involved in two vehicle front-to-side

**Table 2** MAIS 3–6 injured occupants involved in two passenger vehicle front-to-side crashes

| Crash configuration | Injury risk ratio of MAIS 3–6 injured occupants:drivers (striking vehicle/struck vehicle) |
|---|---|
| PC to PC | 1 (2088):2.09 (4354) |
| | 1 (2072):2.05 (4242) |
| PC to LTV | 2.06 (667):1 (324) |
| | 2.77 (653):1 (236) |
| LTV to PC | 1 (1573):1.84 (2897) |
| | 1 (1360):2.13 (2892) |
| LTV to LTV | 1 (1398):1.49 (2081) |
| | 1 (1185):1.32(1565) |
| Total | 1 (5726):1.69 (9656) |
| | 1 (5270):1.70 (8935) |

LTV, light truck vehicles; MAIS, maximum Abbreviated Injury Score; PC, passenger cars.

**Table 3** Distribution of maximum Abbreviated Injury Score (MAIS) 3–6 injured occupants by body regions and lateral location of damage in USA in 2007 (N=564, missing data=141)

| Lateral location of damage (95% CI) | 1 | 2 | 3 | 4 | 5 | 6 | 7 | 8 | 9 |
|---|---|---|---|---|---|---|---|---|---|
| **Occupants number** | | | | | | | | | |
| Percentage | 1.6 (0.56 to 2.64) | 20.39 (19.39 to 21.43) | 27.48 (26.44 to 28.52) | 31.03 (29.99 to 32.07) | 14.54 (13.5 to 15.58) | 4.96 (3.92 to 6) | 0 (0 to 1.04) | 0 (0 to 1.04) | 0 (0 to 1.04) |
| Cumulative percentage | 1.6 | 21.99 | 49.47 | 80.5 | 95.04 | 100 | 100 | 100 | 100 |
| **Five main body regions** | | | | | | | | | |
| Head/face/neck | 3.47% | 23.56% | 20.67% | 36.2% | 14.94% | 1.17% | 0 | 0 | 0 |
|  | 1.07 (0.84 to 1.3) | 7.27 (7.04 to 7.5) | 6.38 (6.15 to 6.61) | 11.17 (10.94 to 11.4) | 4.61 (4.38 to 4.84) | 0.36 (0.13 to 0.59) | 0 (0 to 0.23) | 0 (0 to 0.23) | 0 (0 to 0.23) |
| Chest | 0 | 18.89% | 23.07% | 26.58% | 23.78% | 7.69% | 0 | 0 | 0 |
|  | 0 (0 to 0.23) | 4.79 (4.56 to 5.02) | 5.85 (5.62 to 6.08) | 6.74 (6.51 to 6.97) | 6.03 (5.8 to 6.26) | 1.95 (1.72 to 2.18) | 0 (0 to 0.23) | 0 (0 to 0.23) | 0 (0 to 0.23) |
| Pelvis | 0 | 25% | 45.51% | 22.69% | 4.49% | 2.31% | 0 | 0 | 0 |
|  | 0 (0 to 0.23) | 1.95 (1.72 to 2.18) | 3.55 (3.32 to 3.78) | 1.77 (1.54 to 2) | 0.35 (0.12 to 0.58) | 0.18 (0 to 0.41) | 0 (0 to 0.23) | 0 (0 to 0.23) | 0 (0 to 0.23) |
| Upper extremity | 2.12% | 18.26% | 34.41% | 31.19% | 9.71% | 4.31% | 0 | 0 | 0 |
|  | 0.35 (0.12 to 0.58) | 3.01 (2.78 to 3.24) | 5.67 (5.44 to 5.9) | 5.14 (4.91 to 5.37) | 1.6 (1.37 to 1.83) | 0.71 (0.48 to 0.94) | 0 (0 to 0.23) | 0 (0 to 0.23) | 0 (0 to 0.23) |
| Lower extremity | 0.93% | 17.43% | 30.26% | 32.13% | 10.09% | 9.16% | 0 | 0 | 0 |
|  | 0.18 (0 to 0.41) | 3.37 (3.14 to 3.6) | 5.85 (5.62 to 6.08) | 6.21 (5.98 to 6.44) | 1.95 (1.72 to 2.18) | 1.77 (1.54 to 2) | 0 (0 to 0.23) | 0 (0 to 0.23) | 0 (0 to 0.23) |

crashes. The highest risk ratio of MAIS 3–6 injuries was 1 vs 2.09, which resulted from PC-to-PC side impact crashes. Comparatively speaking, the LTV-to-LTV side impact crashes caused the lowest injury risk ratio for MAIS 3–6 injured occupants.

### Specific lateral location of damage

Approximately 31% (95% CI 29.99% to 32.07%) of MAIS 3–6 injuries covered the extent zone 4 of damage. Zones 2 and 3 comprised only 20.4% (95% CI 19.39% to 21.43%) and 27.5% (95% CI 26.44% to 28.52%) of MAIS 3–6 injuries (table 3). As can be seen, the crush zones attributed to occupants seriously injured were zones 2, 3 and 4. The 50th centile of MAIS 3–6 injuries occurred when the extent of damage crush was in zone 3 of the struck vehicle.

The information in table 3 shows that the majority of AIS 3–6 injuries to the head/face/neck (36.2%), chest (26.58%) and lower extremity (32.13%) were associated with an extent of damage in zone 4, while the pelvis (45.51%) and upper extremity (34.41%) were associated with an extent of damage in zone 3. The extent of damage in zones 3 and 4 accounted for about 56.87% of head/face/neck injuries, about 49.65% of chest injuries, about 68.2% of pelvis injuries, about 65.6% of upper extremity injuries and about 62.39% of lower extremity injuries, respectively.

Assuming a PC's width of 1.8 m in LTV-to-PC nearside impact crashes, the extent of damage in each was 20 cm. Zone 1 demonstrated that crush intrusion in the lateral location of damage was between 0 and 20 cm, zone 2 demonstrated that crush intrusion was between 20 and 40 cmKzone 9 demonstrated that crush intrusion was between 160 and 180 cm and included broken crush, according to the J224 Collision Deformation Classification of SAE Surface Vehicle Standard. However, the usual distance between occupants and the interior panel is less than 20 cm. Therefore, the crush intrusion of nearside struck vehicle, if less than 60 cm, will not cause the occupants to lose capacity or fatal injury. In the current study, the risk ratio of MAIS 3–6 injuries was 80.5%, resulting from the extent of damage extending into zone 4.

### Specific horizontal location of damage

The two highest injury ratios of MAIS 3–6 injuries were 25.39% (95% CI 24.46% to 26.32%) and 36.88% (95% CI 35.95% to 37.81%), respectively, and covered the damage locations 'D' and 'Y' (table 4); the lowest injury ratio of MAIS 3–6 injuries was only 5.67% (95% CI 4.74% to 6.6%) and covered the damage location 'B'.

Over half of all body AIS 3–6 injuries were contributed to by the damage locations 'D' and 'Y'. The highest risk ratios of MAIS 3–6 injuries to the head/face/neck, chest, pelvis, upper extremity and lower extremity were about 35.95%, 36.99%, 25.98%, 40% and 47.88%, respectively, associated with the damage zones 'Y' 'D'. In contrast, the lowest risk ratios of MAIS 3–6 injuries to the head/face/neck, chest, pelvis, upper extremity and

**Table 4** Distribution of maximum Abbreviated Injury Score (MAIS) 3–6 injured occupants by body regions and horizontal location of damage in the USA in 2007 (N=705)

| Horizontal location of damage (95% CI) | Unknown | B | D | F | P | Y | Z |
|---|---|---|---|---|---|---|---|
| Occupants number | | | | | | | |
| Percentage | 0.14 (0 to 1.07) | 5.67 (4.74 to 6.6) | 25.39 (24.46 to 26.32) | 10.35 (9.42 to 11.28) | 8.94 (8.01 to 9.87) | 36.88 (35.95 to 37.81) | 12.62 (11.69 to 13.55) |
| Five main body regions | | | | | | | |
| Head/face/neck | 0 | 7.79% | 22.97% | 12.98% | 6.48% | 35.95% | 13.84% |
| | 0 (0 to 0.22) | 2.55 (2.33 to 2.77) | 7.52 (7.3 to 7.74) | 4.25 (4.03 to 4.47) | 2.12 (1.9 to 2.34) | 11.77 (11.55 to 11.99) | 4.53 (4.31 to 4.75) |
| Chest | 0.57% | 4.07% | 36.99% | 6.92% | 9.82% | 30.63% | 11% |
| | 0.14 (0 to 0.36) | 1 (0.78 to 1.22) | 9.08 (8.86 to 9.3) | 1.7 (1.48 to 1.92) | 2.41 (2.19 to 2.63) | 7.52 (7.3 to 7.74) | 2.7 (2.48 to 2.92) |
| Pelvis | 0 | 9.27% | 16.71% | 20.37% | 3.66% | 25.98% | 24.02% |
| | 0 (0 to 0.22) | 0.71 (0.49 to 0.93) | 1.28 (1.06 to 1.5) | 1.56 (1.34 to 1.78) | 0.28 (0.06 to 0.5) | 1.99 (1.77 to 2.21) | 1.84 (1.62 to 2.06) |
| Upper extremity | 0 | 2.64% | 20% | 11.29% | 10.43% | 40% | 15.64% |
| | 0 (0 to 0.22) | 0.43 (0.21 to 0.65) | 3.26 (3.04 to 3.48) | 1.84 (1.62 to 2.06) | 1.7 (1.48 to 1.92) | 6.52 (6.3 to 6.74) | 2.55 (2.33 to 2.77) |
| Lower extremity | 0.48% | 5.3% | 22.82% | 5.3% | 12.91% | 47.88% | 5.3% |
| | 0.09 (0 to 0.31) | 0.99 (0.77 to 1.21) | 4.26 (4.04 to 4.48) | 0.99 (0.77 to 1.21) | 2.41 (2.19 to 2.63) | 8.94 (8.72 to 9.16) | 0.99 (0.77 to 1.21) |

lower extremity were only about 6.48%, 4.07%, 3.66%, 2.64% and 5.3%, respectively, associated with the damage zone 'B'.

## Specific vertical location of damage

Approximately 57.6% (95% CI 55.57% to 59.61%) of MAIS 3–6 injuries resulted from the damage location 'E' (table 5); zone 'G+M' contributed to only 0.14% (95% CI 0% to 2.16%) of MAIS 3–6 injuries.

As can be seen, the majority of all MAIS 3–6 injuries were contributed to by the damage location 'E'; the risk ratios of MAIS 3–6 injuries to the head/face/neck, chest, pelvis, upper extremity and lower extremity were about 55.4%, 50.26%, 70.37%, 71.26% and 60.3%, respectively.

## DISCUSSION

We evaluated the risks of MAIS 3–6 injuries to the head/face/neck, chest, pelvis, upper extremity and lower extremity from two passenger vehicle front-to-side impact crashes. Our analysis was based on an examination of approximately 1 362 000 weighted cases of PC and LTV, which were extracted from the US-NASS/CDS 2007 crash investigations database.

Previous studies have shown that occupants of PC were more likely than occupants of LTV to be seriously injured. About 2.9% of PC occupants and 1.9% of LTV occupants in side-impacted vehicles were seriously injured.[10] The Crash Aggressivity Index of LTV, compared with the reference case PC (0.00), increased for vans (0.32) and medium trucks (0.48), and then for heavy trucks (0.40) and truck tractors (0.21) it decreased.[11] PC–LTV incompatibility in vehicle mass, vehicle geometry and structure also have been widely discussed.[12–14]

In our study, the injuries risk ratio in front-to-side impact crashes was such that overall occupants of struck vehicles had a 69% higher MAIS 3–6 injury risk than overall occupants of the striking vehicle. The reason is that the striking vehicle occupants are, to a greater extent, protected by the presence of a larger structural crush zone, that is, including the energy-absorbing bumper and front-end structure, the vehicle's engine, the front suspension and wheels, the engine-mounting frame and integral firewall, supplemental safety systems. However, the struck vehicle crush zone is comprised of only the side doors and the relatively light framework of the occupant cell, plus interior foams which offer some limited protection, or some vehicle types are now also equipped with supplemental side airbags or curtains which have increased the level of nearside impact protection available to the occupants, but the level of protection does not reach the level which currently exists for frontal crashes.

The results from lateral damage location suggest that strategies and methodologies of occupant protection and injuries reduction are developed from the extent of

**Table 5** Distribution of maximum Abbreviated Injury Score (MAIS) 3–6 injured occupants by body regions and vertical location of damage in the USA in 2007 (N=705)

| Vertical location of damage (95% CI) | Unknown | A | E | G+M | L |
|---|---|---|---|---|---|
| Occupants number |  |  |  |  |  |
| Percentage | 4.82 (2.8 to 6.84) | 37.45 (35.43 to 39.47) | 57.59 (55.57 to 59.61) | 0.14 (0 to 2.16) | 0 (0 to 2.08) |
| Five main body regions |  |  |  |  |  |
| Head/face/neck | 6.5% | 38.1% | 55.4% | 0 | 0 |
|  | 2.13 (1.72 to 2.54) | 12.48 (12.07 to 12.89) | 18.15 (17.74 to 18.56) | 0 (0 to 0.41) | 0 (0 to 0.43) |
| Chest | 4.07% | 50.26% | 45.09% | 0.57% | 0 |
|  | 1 (0.57 to 1.43) | 12.34 (11.91 to 12.77) | 11.07 (10.64 to 11.5) | 0.14 (0 to 0.57) | 0 (0 to 0.43) |
| Pelvis | 7.44% | 22.19% | 70.37% | 0 | 0 |
|  | 0.57 (0.14 to 1) | 1.7 (1.27 to 2.13) | 5.39 (4.96 to 5.82) | 0 (0 to 0.43) | 0 (0 to 0.43) |
| Upper extremity | 2.63% | 26.1% | 71.26% | 0 | 0 |
|  | 0.43 (0 to 0.86) | 4.26 (3.83 to 4.69) | 11.63 (11.2 to 12.06) | 0 (0 to 0.43) | 0 (0 to 0.43) |
| Lower extremity | 3.82% | 35.88% | 60.3% | 0 | 0 |
|  | 0.71 (0.28 to 1.14) | 6.67 (6.24 to 7.1) | 11.21 (10.78 to 11.64) | 0 (0 to 0.43) | 0 (0 to 0.43) |

damage in zones 1, 2 and 3. Two methods are often used to improve vehicle safety in nearside impact crashes. Method number one is to enhance the structural rigidity of the interior panel of side doors to reduce the magnitude of door panel crush, such as the optimised design of the impact bar, foam systems and door interior panels. Method number two is to add the airbags between the occupant and the interior panel, that is, supplemental head curtains and chest airbags, which have increased the level of protection available to occupants. Tencer et al[6] showed that the magnitude of door crush in a nearside impact from category 1 (3–8 cm of crush) to category 2 (9–15 cm of crush) was associated with an average of a 0.4 increase in thoracic AIS, a 0.3 increase in abdominal AIS and a 0.2 increase in the pelvis ($p<0.0001$). In our data, we did not consider crush intrusion on MAIS below 3. We analysed the magnitude of side crush intrusion in a nearside impact from zone 2 (about 20–40 cm of crush) to zone 3 (about 40–60 cm of crush), which was associated with an average of a 0.22 increase in chest MAIS 3–6, a 0.82 increase in pelvis MAIS 3–6 and a 0.88 increase in upper extremity MAIS 3–6 ($p<0.0001$).

The results from horizontal damage location indicate that the crushed region of the impacted vehicle is located mainly in forward of C-pillar in real-world nearside impact crashes; there are only a few possibilities in rearward of C-pillar.

The results from vertical damage location indicate that the crushed region of the impacted vehicle is located mainly between upward of the rocker rail and downward of the windowsill in real-world nearside impact crashes; there are only a few possibilities in upward of the windowsill. Therefore, for design researches, it is useful to enhance the structural rigidity of side doors and rocker rail of PC and consider vehicle front-end aggressively especially accounting for LTV, aim to reduce the occupant's injury risk based on motor vehicle safety standards.[15]

In conclusion, over half of crash types in the real world are attributed to vehicle-to-vehicle crashes. Rigidity of the side doors, A-pillar, B-pillar and rocker rail are priorities for improving the safety level of nearside impact. The specific location of damage that accounted for MAIS 3–6 injured occupants and 50% injury risk is extent zone 3, zones 'Y' and 'E' from the lateral, horizontal and vertical directions of the struck vehicle, respectively. To reduce the risk of death and serious injury, protection of the head, chest and pelvis should be priorities for injury countermeasure development, because these three body regions accounted for over half of the AIS 3–6 injuries suffered from vertical crush zone 'E' in PC–LTV nearside impact crashes.

**Author affiliations**
[1]School of Mechanical and Automotive Engineering, Xiamen University of Technology, Xiamen, Fujian, China
[2]State Key Laboratory of Advanced Design and Manufacture for Vehicle Body, Hunan University, Changsha, Hunan, China
[3]National Crash Analysis Centre, The George Washington University, Ashburn, Virginia, USA

**Acknowledgements** The authors thank Dr Prodeep Mohan (senior scientist of the National Crash Analysis Center, George Washington University), Ms Christina (master graduate student of the National Crash Analysis Center, George Washington University) and Mr Leyu Wang (doctoral graduate student of the National Crash Analysis Center, George Washington University) for their very valuable suggestions and software processing for this study.

**Contributors** YT was the primary investigator. He initiated and designed the study and completed the data processing and manuscript writing. LC and SK participated in developing the study concept and data interpretation. All authors reviewed and approved the final manuscript.

**Funding** This project was supported by the National Crash Analysis Center (NCAC), the George Washington University (GWU) and sponsored by the National Natural Science Foundation of China, grant no.51305374 and no.51355001. Open Foundation of State Key Laboratory of Advanced Design and Manufacture for Vehicle Body of Hunan University, grant no. 31215004. Fujian Natural Science Foundation of China, grant no. 2012J05103.

**Competing interests** None.

**Provenance and peer review** Not commissioned; externally peer reviewed.

**Data sharing statement** The Distribution of MAIS 3–6 injured occupants by body regions and lateral location of damage in USA in 2007 is available by contacting the corresponding author.

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
