## [Reviewer comments · BMJ Open]

Some articles will have been accepted based in part or entirely on reviews undertaken for other BMJ Group journals. These will be reproduced where possible.

ARTICLE DETAILS

TITLE (PROVISIONAL)	Associations between damage location and five main body regions injuries of MAIS 3-6 injured occupants
AUTHORS	Tang, Youming; Cao, Libo; Kan, Steven

VERSION 1 - REVIEW

REVIEWER	Dr Mike Bambach TARS, University of New South Wales, Australia
REVIEW RETURNED	13-Jan-2014

GENERAL COMMENTS	The authors identify front-to-side crashes which includes PC-PC and LTV-LTV crashes (table 2), yet there is no indication in the methods if the results in tables 3-5 are for all crashes or only PC-LTV and/or LTV-PC crashes. Please clarify and provide the case counts for the PC-LTV study. Results para 1: it seems the results in table 1 are for all occupants, not injured occupants, in which case the stated injury rates in the text would not be correct - please clarify Results para 4: should table 1 be table 3? Results para 6: should the 85% refer to zone 4 not zone 3? Discussion para 2: where the authors discuss results of other studies of PC-LTV side crashes it would be useful to discuss and compare the results of this study
--

REVIEWER	Peng Zhang International Center for Automotive Medicine University of Michigan USA
REVIEW RETURNED	29-Jan-2014

GENERAL COMMENTS	Yes and I have performed this review This is a well-thought study, which tries to investigate the association between damage locations and MAIS 3+ injuries in front-to-side crashes. Better understanding of it can lead to improvement of car design in protecting occupants. Overall, it is well-written. Minor issues: The authors have discussed more on passenger cars (PC) – light truck vehicles (LTV) crashes in their discussion, and also in table 2. Clearly, that is one of their considered points in the design of their
--

	study. I don't know why they went away from it in the later analysis (table 3-5). Major issues: It might be better to get a statistician involved to improve upon the analysis.  1. Need confidence intervals other than just percentage, so one knows how significant differences are among different groups. 2. Didn't consider the confounding variables, such as occupants' factors and crash factors. I think at least near- or far- side impact should be considered, since the authors considered side impacts. Injury risks are very different for near vs far side crashes.
--	--

VERSION 1 – AUTHOR RESPONSE

Reviewer 1:

Comment 1: The authors identify front-to-side crashes which includes PC-PC and LTV-LTV crashes (table 2), yet there is no indication in the methods if the results in tables 3-5 are for all crashes or only PC-LTV and/or LTV-PC crashes. Please clarify and provide the case counts for the PC-LTV study.

Response 1: Thanks for the reviewer's kind suggestion. We have added the number of all near-side MAIS 3-6 injured occupants seated directly adjacent to the struck-side of the vehicle, in side impact crashes which include LTV-LTV and PC-PC. The Number can be found in table 3-5 title.

Comment 2: results para 1: it seems the results in table 1 are for all occupants, not injured occupants, in which case the stated injury rates in the text would not be correct - please clarify

Response 2: Thanks for the reviewer's kind suggestion. We had amended, and stated injury risk rates for all occupants using the words 'injuries probability'.

Comment 3: Results para 4: should table 1 be table 3?

Response 3: Thanks for the reviewer's suggestion. We have checked and modified to table 3.

Comment 4: Results para 6: should the 85% refer to zone 4 not zone 3?

Response 4: Thanks for the reviewer's suggestion. We have checked and modified to zone 4.

Comment 5: discussion para 2: where the authors discuss results of other studies of PC-LTV side crashes it would be useful to discuss and compare the results of this study.

Response 5: Thanks for the reviewer's kind suggestion. We have added sections to discuss and compare the crush intrusion in near-side impact which affected the chest and pelvis MAIS from results of Tencer et.al (2005) studies to our finding. The relevant sections can be found in Discussion paragraph 4.

Reviewer 2:

Comment 1: The authors have discussed more on passenger cars (PC) – light truck vehicles (LTV) crashes in their discussion, and also in table 2. Clearly, that is one of their considered points in the design of their study. I don't know why they went away from it in the later analysis (table 3-5).

Response 1: Thanks for the reviewer's comment. In our study, for readers, table 2 was shown an overview of MAIS 3-6 injured occupants selected from NASS/CDS for data analysis, the front-to-side impact accounted for about 9656 of MAIS 3-6 injured occupants and 8935 of MAIS 3-6 injured drivers involved in struck vehicle, 5726 of MAIS 3-6 injured occupants and 5270 of MAIS 3-6 injured drivers involved in striking vehicle, respectively. The later injury analysis (table 3-5) was conducted base those data of near-side occupants of struck vehicle which retrieved from table 2 by Selection criteria, in a LTV-PC side impact crashes.

Comment 2: Need confidence intervals other than just percentage, so one knows how significant differences are among different groups.

Response 2: Thanks for the reviewer's kind suggestion. According to his advices, 95% confidence interval (C.I.) was used to descript the probability of injury risk. The revised details can be found in table 3-5, ABSTRACT (Results), and paragraph 1 of Specific lateral location of damage, Specific horizontal location of damage and Specific vertical location of damage.

Comment 3: Didn't consider the confounding variables, such as occupants' factors and crash factors. I think at least near- or far- side impact should be considered, since the authors considered side impacts. Injury risks are very different for near vs far side crashes.

Response 3: Thanks for the reviewer's kind suggestion. According to his advices, injury analysis was adjusted to conduct based on those injuries sustained by the near-side occupants seated directly adjacent to the struck-side of the vehicle, in a LTV-PC side impact crashes. The relevant sections can be found in Selection criteria, the data of final results can be found and updated in Results paragraph 3-4, 6-9. Actually we are doing some research about crash factors, such as Delta-V (striking vehicle and struck vehicle) associated with five main body regions injuries of MAIS 3-6 injured occupants, in a LTV-PC side impact crashes.